# Innovative Structural Fuse Systems for Various Prototype Applications

**DOI:** 10.3390/ma15030805

**Published:** 2022-01-21

**Authors:** Alireza Farzampour

**Affiliations:** Department of Permitting, Inspections and Enforcement, Washington, DC 20774, USA; afarzam@vt.edu

**Keywords:** dampers, cyclic response, energy efficiency, ductility

## Abstract

To resist the imposed lateral forces on the structures, hysteric dampers are developed from steel plates and strategically implemented within various structural applications. Structural shear dampers have recently been used to alleviate damage, while remaining members remain intact and undamaged. The practical use of the innovative dampers in structural applications is investigated in this study. For this purpose, the design methodology for a set of innovative shear dampers is initially elaborated, for which the dampers are designed considering the governing shear and flexural ductile limit states, while the brittle buckling limit state is prevented. Subsequently, the finite element modeling methodology is verified and compared to laboratory tests for computationally analyzing various shapes of the shear damper in structural applications. Three major general prototype structures are established, and shear dampers are designed to be incorporated in prototype applications. For each of the proposed applications, at least six different shapes of shear dampers are designed and subsequently compared with conventional systems. The results determined that the use of innovative shear dampers could effectively reduce demand forces on the boundary elements by more than 40% on average, while the strength and the stiffness alter within margin of difference less than 5%.

## 1. Introduction

Structures are designed to prevent collapse and limit the damage under severe earthquakes. For limiting the damage to a structural system, the ductile behavior of steel structural shear dampers is considered based on the inelastic drift capacity and desired energy dissipation under excessive loading conditions. These structural fuse systems are used to protect the surrounding elements from inelastic damage, while concentrating the inelasticity in a specific of the buildings. One class of structural steel fuse with high energy dissipation capacity is made of steel plates with shear links cut into them, to initiate yielding ductile modes of behavior as the plate undergoes the shear loading demand [1].

The structural fuses are intended to be accessible, effective, and replaceable [2,3,4,5]. By the implementation of the structural dampers, the surrounding elements are protected from serious damage, and the dampers can perform efficiently after being replaced. The structural shear dampers can be implemented in various applications (Figure 1). Figure 1 shows schematic examples of structural shear damper implementation for various applications. The structural shear links are developed to improve various system issues, for which several shapes are represented in Figure 1g.

Several studies have indicated that strategic dampers are able to be incorporated in different applications [6,7,8,9]. Previous works showed that using the slit panels, butterfly-shaped dampers, or hysteretic structural fuses with different shapes could effectively reduce the response of multistory structures under earthquake loading condition [3]. To protect the beam–column connection from considerable plastic strain concentration, Oh et al. [10] used butterfly-shaped dampers, which effectively saved the beam from the accumulation of plastic strain under large deformations. Luth et al. [11], and Shin et al. [12] used hourglass shape fuses within the web of the beam to promote the yielding mechanism as the dominant limit state under cyclic loading condition [13,14,15,16]. It is shown that straight links in steel plate shear walls with an appropriate aspect ratio increase the ductility of the whole system.

The structural steel fuses are designed to be replaceable after the occurrence of major events. Therefore, they decrease the costs by a significant margin. The cost efficiency of using replaceable dampers is related to less damage to the surrounding members, faster return to performance level, and lower demand on columns and beams [17,18,19,20,21,22,23,24,25,26,27,28,29,30]. Several studies have investigated and addressed the common problems associated with structural fuse systems consisting various dampers [31,32,33,34,35,36]. Ma et al. [2] concluded that butterfly-shaped dampers are efficient by aligning the demand moment curves with capacity moment diagrams. Several works proposed a design methodology with which the butterfly-shaped and straight dampers are able to effectively resist lateral torsional buckling. Along the same lines, previous works [14,15,16] showed that the usage of fuses, if appropriately designed, leads to a proper distribution of combined shear and flexural stresses within the length damper [14]. Along the same lines, various computational approaches were previously suggested to study the behavior of structural systems [32,33,34]. 

In this study, the effects of fuses with different geometrical configurations are investigated with finite element analysis. Based on the applications studied in Structural Engineers Association of California (SEAOC) [34,35] examples with typical solid steel plate, three distinctive groups are selected, and fuses are designed for each set. The design is conducted based on achieving the same strength for all the fuses within each prototype application group. The investigated output parameters are selected based on the previous studies, which are mainly associated with energy dissipation capability, ductility, strength, stiffness, and fracture prevention. Subsequently, the appropriate structural damper system for each group is identified and the implementation advantages are elaborated. The limitation of this study is related to replaceability issues and system performance in the case extreme events for which the replaceable dampers do not fit into the system due to the occurrence of several damages. In addition, the cost of preparation and complicated design requirements could be considered as further limitations for such systems.

## 2. Discussion for Different Shapes for Structural Fuses

### 2.1. Uniform Yielding Design Concept

To effectively design structural dampers, one of the main procedures reported previously in the literature is to have a uniform yielding over the length of the link [2], indicating simultaneous ductile yielding limit state occurrence along the length of the seismic shear damper. Figure 2 shows the schematic hourglass shaped dampers, and the major and minor bending axes of x and y. Figure 3 presents the moment diagrams resulting in strategic plastic hinges, and the demand force related to a typical damper system. It is recommended to have the hinges developed far from sharp geometrically points and to control the strain concentration over joints. The global shear buckling transform into more ductile mechanisms to have desirable structural performance [2,3,4]. Figure 3 shows the demand forces determined in Equations (1) and (2).
(1)$$M_{p}=\frac{w{\left(z\right)}^{2}t}{4}\sigma_{y}$$
(2)$$M_{\mathrm{loading}}=Pz$$
where t and *w*(*z*) are the thickness and width; M_loading_ indicates the demand; M_p_ indicates the capacity. By equating the force demand and force capacity terms, the width of the shear link varying along the length could be derived:(3)$$w\left(z\right)=\sqrt{\frac{4Pz}{t\sigma_{y}}}$$

Therefore, *b* determining the end width, is achieved based on Equation (4):(4)$$b=\sqrt{\frac{2PL}{t\sigma_{y}}}$$

The width of the link at each section is indicated based on Equation (5)
(5)$$\left(w\left(z\right)/b\right)=\sqrt{\left(\frac{z}{L}\right)}$$

The proposed shape shown in Figure 4 determines a desirable solution for having uniform yielding condition and avoiding the lateral torsional buckling limit state. 

### 2.2. Uniform Curvature Design Concept

The uniform curvature design concept considered the uniform curvature method for reestablishing various efficient geometrical shapes for in-plane structural dampers based on previous experimentation [36]. It is suggested that seismic shear fuses with the uniform curvature would spread the plasticity along the length of the link to avoid the concentration of the plastic strains. 

To enjoy the benefits of the same curvature for having subsequent yielding conditions along the length of the link, it is required to set the curvature, as shown in Figure 5 and summarized in Equation (6).
(6)$$\phi =\frac{M}{EI}=\frac{Pz}{EI}=\frac{Pz}{E\left(\frac{b_{z}^{3}t}{12}\right)}$$

After simplification, the width could be established as follows:(7)$$b_{z}=\sqrt[{3}]{\left(\frac{12Pz}{E\phi t}\right)}\;$$

Similarly, the end width could be established by Equation (8)
(8)$$b=\sqrt[{3}]{\left(\frac{6PL}{E\phi t}\right)}\;$$

Therefore, Equation (9) is obtained.
(9)$$\frac{w\left(z\right)}{b}=\sqrt[{3}]{\frac{2z}{L}}\;$$

Based on Equations (5) and (9), the uniform yielding along the damper length occurs if the width of the hourglass-shaped damper, *w*(*z*), corresponds to the square root of z. Along the same lines, the constant curvature occurs if the width aligns with the cube root of z which is schematically shown in Figure 6. 

### 2.3. Limit States and Ductile Based Design Concept

The elaborated guideline in this section is proposed to determine the procedures to design various dampers with different shapes, as shown in Figure 7 for use in seismic structural fuses [14,15]. 

The provided guidelines improve the energy dissipation capability of the shear dampers by having the flexural modes as the dominant mode of behavior for ductile performance. It is shown that butterfly-shaped dampers with the dominant flexural mode of behavior are able to dissipate energy efficiently with full hysteretic behavior while the plastic strain concentration areas would occur far from the sharper geometrical changes, which leads to a lower probability of fracture protentional [15].

The transitional equations are obtained as follows (For *a* < *b*/2). For having flexure limit states governed over the shear, the geometry should satisfy the following Equations (3)–(6):(10)(b−a)/L<0.28  (or α>148°)         Flexure dominated 

In addition, the strength limit states are as follows:(11)$$P_{p}^{flexure}=\frac{2n\left(b-a\right)at\sigma_{y}}{L}$$
(12)$$P_{p}^{shear}=n\frac{\sigma_{y}at}{\sqrt{3}}$$

For *a* > *b*/2 or slit (*a* = *b*):(13)b/L<1.15          Flexure dominated

The shear and flexure capacity are summarized as follows:(14)$$P_{p}^{flexure}=\frac{nb^{2}t}{2L}\sigma_{y}$$
(15)$$P_{p}^{shear}=n\frac{\sigma_{y}bt}{\sqrt{3}}$$

### 2.4. Brittle Modse, and Developing the Ductile Limit State 

The lateral torsional buckling brittle limit state, flexure, and shear limits state are the most common limit states related to butterfly-shaped dampers [2,3] and the minimum values should be considered according to Equation (16). Based on the parametric study [4,6], the over strength factor (W) for different shapes is derived and summarized in Table 1 [15]. By satisfying Equation (18), the buckling limit state shown in Equation (17) is prevented while the governing mode of behavior would be ductile shear or flexural limit state.
(16)$$P^{p}=\min\left\{P_{p}^{flexure},P_{p}^{shear}\;\right\}$$
(17)$$PLTB_{cr}=\frac{2E[0.533+0.547\left(a/b\right)-0.281{\left(a/b\right)}^{2}+0.096\left(a/b{)}^{3}\right]bt^{3}}{L^{2}\sqrt{1+v}}$$
(18)$$\Omega P^{p}<P_{cr}{}^{LTB}$$

It is shown that the oval-shaped dampers are designed based on the uniform yielding concept, while circular-shaped dampers are designed based on the critical section at the middle of the shear damper. For slits or straight dampers, shear and flexural butterfly-shaped shear links are established according to proposed guidelines elaborated in this study. 

## 3. Designing Prototype Buildings with Innovative Seismic Shear Dampers

### 3.1. Single Row of Links (SRLs)

The Structural Engineers Association of California (SEAOC) prototype application for eccentrically braced systems (EBF) [34,35] is considered and redesigned with innovative seismic structural shear dampers. The design procedures are elaborated in previous sections [34,35]. 

The design for the first prototype system is established for the force of 530 kN based on the reported capacity for the EBF system in SEAOC for EBF system. The EBF system is located on the permitted part of the building is shown in Figure 8a,b. The designed beam is a W10 × 68 with a yielding stress of 322 MPa and 1% hardening. Figure 8 shows the top view and elevation view for which the SRLs are developed.

### 3.2. Design of Single Row of Dampers for EBF System

Six different seismic shear dampers are investigated and compared with the corresponding EBF system. The first set of seismic dampers is flexurally dominated butterfly-shaped ones (FBF), elaborated based on guidelines [15] and shown in Figure 9. The circular (named Circle and shown in Figure 9) and shear dominated butterfly shaped dampers (named SBF and shown in Figure 9) are generally used to accumulate the stresses at the middle section of the seismic damper. The oval shape model (Oval) is developed based on the uniform yielding design concept [15], for which the inelasticity spreads out more uniformly along the length of beam (Figure 9). The straight damper (named straight or slits, and it is shown in Figure 9) is a commonly used straight seismic shear damper. The simple solid plate (named solid and shown in Figure 9) is the conventional shape for which the EBF systems were previously established. Figure 9 shows the designed seismic structural shear dampers. For this purpose, L/60 and L/15 are used as recommended in previous studies for the end angles and the middle [15,34,35].

### 3.3. Design of for MRLs System

Coupled shear wall application based on the SEAOC example is considered and redesigned with innovative seismic shear dampers following the elaborated design methodology [34,35]. Six different seismic shear damper configurations are accordingly considered for this specific application. The geometrical properties used for the mentioned coupling beam systems were determined in SEAOC examples [34,35] and are shown in Figure 10. 

The demand force of 2200 kN and the constitutive material property with steel yielding stress of 322 MPa steel and 1% hardening are considered. Figure 11 shows the designed seismic systems based on the provided guidelines. Flexurally butterfly-shaped dampers are investigated to obtain desirable stress distribution, as shown in Figure 11a. The circular and shear dominated butterfly shaped dampers are represented in Figure 11b,c. The oval shaped link shown in Figure 11d is established based on the uniform curvature concept. The straight damper is a straight shaped geometry used in shear walls and seismic systems to generate the hinges at the ends, as shown in Figure 11e. The simple solid plate, shown in Figure 11f, is the conventional shape. 

### 3.4. Perforated Rows of Links (PRLs)

Various cutout geometries are considered, and specific properties are suggested to study the innovative systems determined in Figure 12. The design is based on the demand force of 1300 kN and material model follows the previous studied computational models. The plan view and elevation view of the conventional lateral resisting system is shown in Figure 12.

The flexurally dominated butterfly-shaped damper is considered to improve the uniformity of stress distribution (Figure 13a). The circular (Figure 13b) and shear dominated butterfly shaped dampers (Figure 13c) are typically used to concentrate the stress within the middle section, while the circle shaped system initiates a better distribution of plastic strains. The oval shape model (Figure 13d) is based on the uniform curvature concept for which the inelasticity spreads along the length of the damper uniformly, and the straight dampers impose the flexural hinge formations at the far ends (Figure 13e). The simple conventional application is shown in Figure 13f. The studied design shapes are summarized in Figure 13.

## 4. Finite Element Modeling Methodology

Finite element software packages [37] are implemented to verify the modeling methodology of seismic shear links. Two laboratory tests are considered for verification purposes [2,12]. The first specimen is B10-36W, adopted by Ma et al. [2] for having lateral torsional buckling and a ductile flexural yielding mechanism, as shown in Figure 14. In this model, the top edge is fixed against all degrees of freedom except horizontal displacement, and the bottom edge is completely restrained against all degrees of freedom. Four-nodded shell element reduced integration (S4R) is implemented with the capability of shear locking and hour glassing resistance, in which five integration points through the thickness are considered. The finest mesh size of 10 mm is chosen for meshing the butterfly shaped shear dampers. To run the analysis, the dynamic explicit solver is used and the AISC 341 [38] cyclic loading protocol for eccentrically braced systems is applied at the unrestrained edge. The constituted material model is applied based on the coupon test reported by Ma et al. [2], in which the material had a yield stress of 273 MPa, ultimate stress of 380 MPa, and linear kinematic hardening between yielding and ultimate stress. The results of verification study for monotonic and cyclic loading are summarized in Figure 15.

The second laboratory test specimen is a beam with circular cutouts along the length of the web section [38]. A twenty-nodded solid element is used to avoid hourglass and shear locking effects. Based on the computational study by Shin et al. [38], a bilinear stress-strain constitutive model is considered with 379 MPa yield stress and elastic modulus of 200 GPa. The story shear is calculated by multiplying the beam shear obtained from FEA analysis by 1.43 and the story drift is estimated by the beam chord rotation divided by 1.43 according to the frame geometry. The chord rotation is monitored from the transverse displacement divided by the clear span of the modeled beam. Figure 16a shows the specimen and the computational model, while Figure 16b shows the verified hysteretic behavior of the experiment and corresponding cyclic pushover curve. 

## 5. Discussion of the Results

The results obtained from the FEA for the use of different seismic innovative shear dampers in various applications are investigated and elaborated in detail. The envelope pushover behavior, plastic strain values, yielding and ultimate strength, over strength, and initial stiffness of each model are derived and compared with other innovative dampers and conventional systems. The modeling methodology precisely follows the verified specimens [2,38]. 

### 5.1. Investigation of Single Rows Seismic Behavior

Figure 17 shows the pushover behavior of the system with seismic links. In general, the fuse system resists the applied loading and subsequently experiences the catenary action up to a specific point at which the buckling or excessive rotations within various elements occur, leading to a large degradation in the stiffness and strength of the systems. The results are estimated and represented in Table 2 and Table 3.

As shown in Figure 17, the results are summarized in Table 2. The equivalent plastic strain at 0.08 drift ratio is monitored for different models to investigate the possibility of brittle effects which is determined in Table 2. 

Considering the plastic strains (PEEQ) FBF, circle and oval shapes could be appropriate options. If the over strength is considered, the circle-shaped and oval-shaped dampers could work efficiently. Figure 18 shows the Von-Mises stress distribution for each damper. The FBF model developed the hinges at the quarter points. The circle- and SBF-shaped models developed the maximum shear stress at the middle portion of the dampers, for which the strain accumulation points are closer to sharper geometrical changes. The straight damper developed the two flexural hinges at the ends. Hence, fracture is more likely to occur in these systems. The oval-shaped system shows fewer strain values due to the constant curvature design. In addition, Table 3 summarizes the performance behavior results.

### 5.2. Investigation of Multiple Rows of Links (MRLs) Results

MRLs are designed for the demand force of 2200 kN [34]. Table 4 and Table 5 show the results related to multiple rows of dampers.

The MRLs rotation at the middle part is highly reduced;. Hence, the local carrying capacity is reduced. It is concluded that the rotation of the middle part prevents the system from each ductile mechanism and the whole system from performing in a desirable fashion or reaching the yielding stage as expected. The backbone curves for the MRLs are summarized in Figure 19. The major decrease in lateral load capacity is determined due to the middle steel plate rotation, and the performance biobehavioral change from the local flexure or shear yielding and subsequent axial yielding. This rotation prevents the dampers from performing in a desirable fashion. 

The behavior of MRLs is not desirable since the desirable mode of behavior did not occur, and the inelasticity was not distributed uniformly. The Von-Mises stresses are determined in Figure 20 for various innovative MRL dampers, indicating that either a stiffener is recommended at the middle to prevent the middle rotation or larger size SRLs could be directly used as proper substitutions.

### 5.3. Investigation of Perimeter Rows of Links (PRLs) Results

The considered designed demand force for PRL computational models is 1300 kN based on the prototype application [34], which is determined in Figure 21 and Table 6. It is concluded that, for the simple shear wall, the field action is generated over lower drift values which is summarized in Table 7. Therefore, the equivalent plastic strain values are lower. Additionally, the simple solid plate system has lower over strength values. However, the boundary element forces are tangibly larger, as shown in Table 8. The tension field action for the first and last floor generates significant demands on boundary elements.

The Von-Mises stress distribution is indicated in Figure 22 at 0.08 drift ratio. It is determined that for various systems with seismic shear dampers, the yielding starts in the damper and plastic strains are concentrated in dampers rather than plates. Additionally, the moments at the middle of the beam and at the middle of the column are estimated as shown in Figure 23. 

Table 8 determines the mid-point demand values for moments to determine the boundary element forces. The tension field action occurred in a simple application, while for the shear walls with innovative dampers, the boundary elements undergo lower demand forces due to a local yielding mechanism produced by dampers. The demand forces on boundary elements for the solid plate system and a system equipped with dampers are considered as the main reason the plate with dampers could be selected as a desirable substitute over the solid plate. It is shown that different innovative dampers could be designed and used to address the high boundary element demand force issues. 

## 6. Conclusions

The structural fuses used in prototype buildings are divided into major three groups. Three groups are considered as follows:Single row of links (e.g., EBF system, linking beam and linked column applications).Multiple rows of links with multiple rows of dampers (e.g., bay bridge application).Perimeter rows of links (e.g., steel shear walls).

The strategic transformation of the behavior mechanism from brittle behavior to local flexure and shear yielding improves the cyclic resistance capability. Based on the results of the different prototype investigations using FE models, methodologies are determined that allow structural tuning. By controlling the buckling limit state, the ductile mechanisms would develop from the flexural and shear stresses. The investigations are summarized as below.

Global shear deformations and the transformation to local flexural yielding mechanism are studied.For steel structural applications, due to the improvement of energy dissipation and reduction of demands on the structural boundary elements, the new generation of innovative damper are suggested for use.Different shapes of innovative dampers could be designed and used to improve the high boundary element demand force issue and reduce the demand forces up to more 40% in average.

The limitations of this study are related to replaceability issues and system performance in the case extreme events for which the replaceable dampers do not fit into the system due to the occurrence of several damages. In addition, the cost of preparation and complicated design requirements could be considered as further limitations. 

## Figures and Tables

**Figure 1 materials-15-00805-f001:**
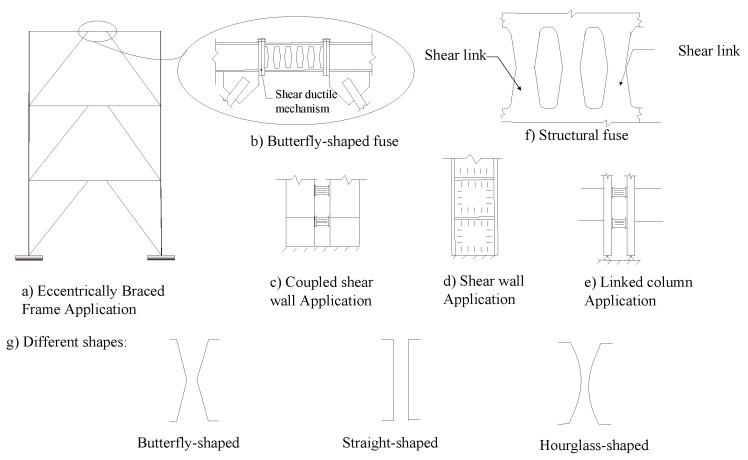
Structural fuse system. (**a**) Eccentrically braced system; (**b**) Butterfly-shaped fuse; (**c**) Coupled shear wall; (**d**) Shear wall; (**e**) Linked column application; (**f**) Structural fuse; (**g**) Different shapes.

**Figure 2 materials-15-00805-f002:**
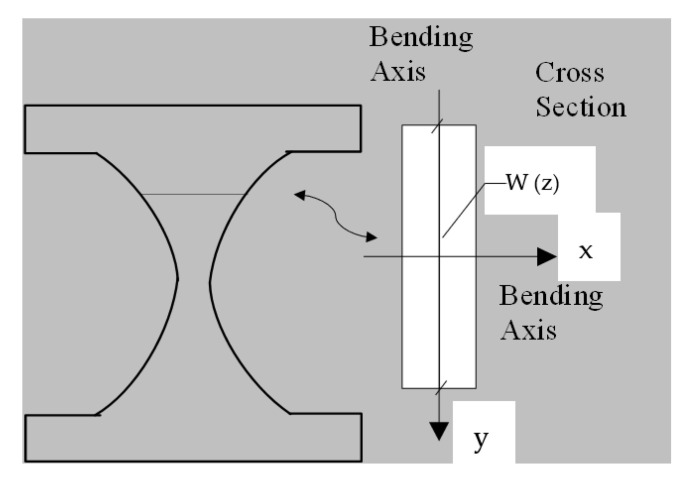
The schematic concept of uniform yielding for the shear damper design.

**Figure 3 materials-15-00805-f003:**
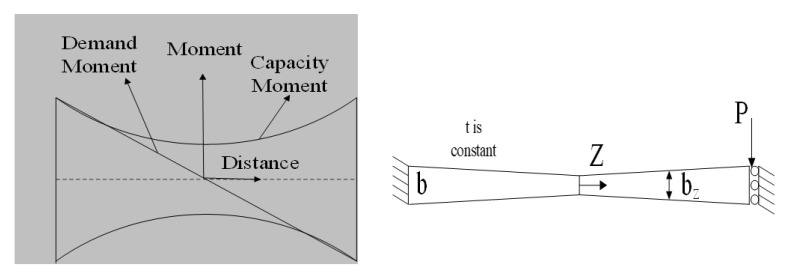
The butterfly shaped shear fuse loading condition.

**Figure 4 materials-15-00805-f004:**
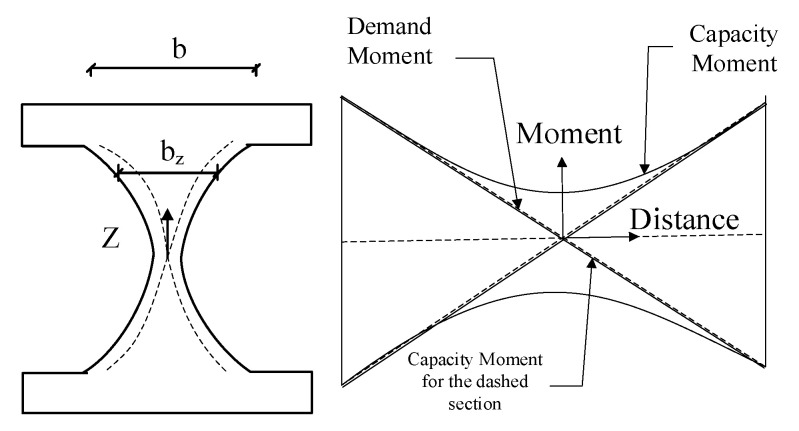
The damper geometry for uniform yielding design concept.

**Figure 5 materials-15-00805-f005:**
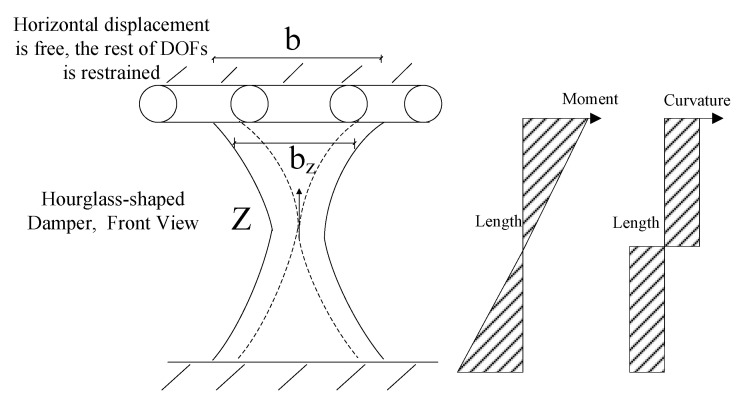
The geometry of the butterfly-shaped damper for constant curvature concept.

**Figure 6 materials-15-00805-f006:**
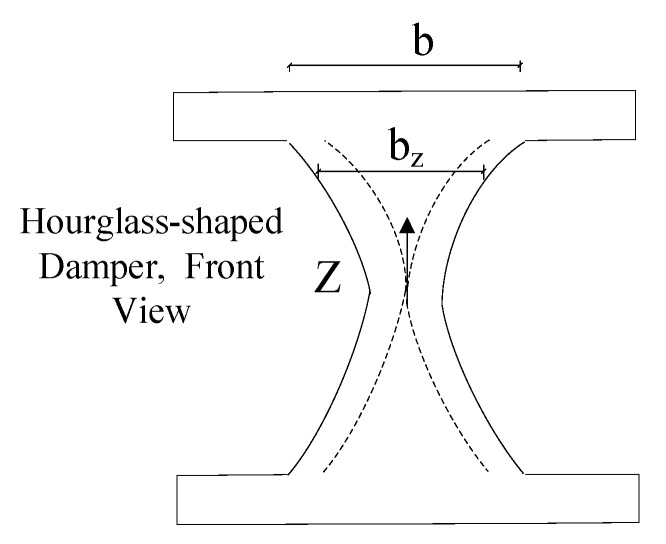
The constant curvature concept.

**Figure 7 materials-15-00805-f007:**
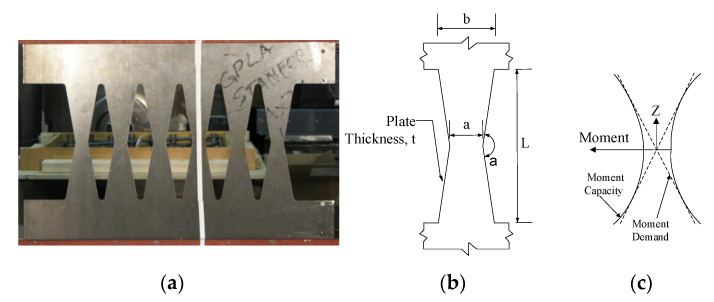
The hysteretic damper general geometrical properties and loading conditions. (**a**) Butterfly-shaped Plate; (**b**) Geometry of a damper; (**c**) Moment Diagram.

**Figure 8 materials-15-00805-f008:**
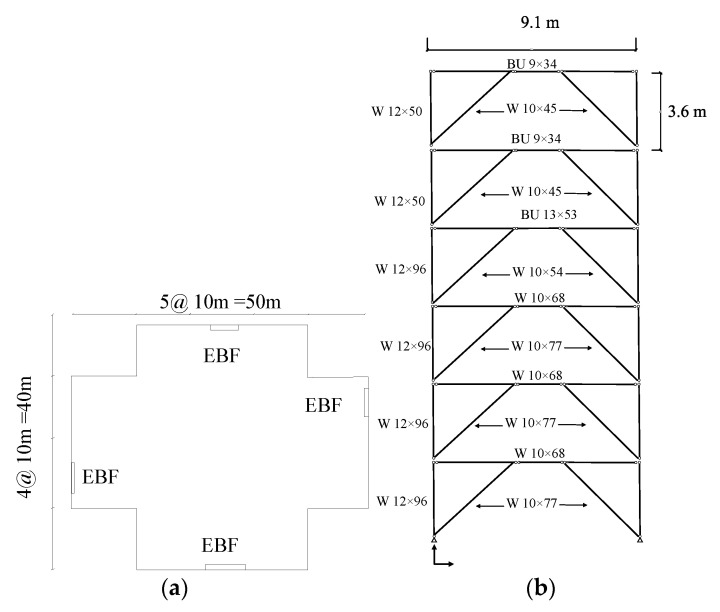
The SEAOC example for EBF system with six stories (**a**) The plan view of the structure with EBF system shown on the perimeter of the building (SEAOC, 2012) (**b**) The six story EBF system with columns, beams and braces sectional properties.

**Figure 9 materials-15-00805-f009:**
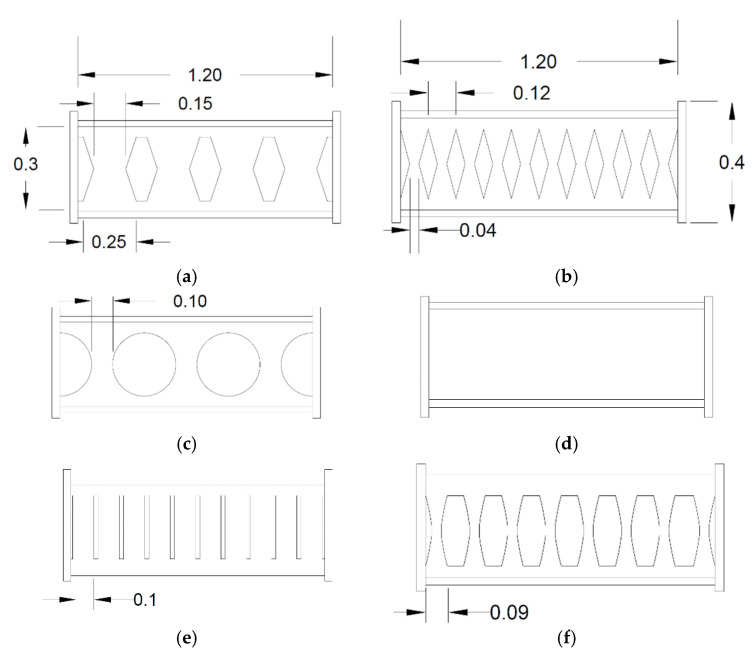
The designed models for EBF system (**a**) FBF (**b**) SBF (**c**) Circle (**d**) Solid (**e**) Straight (**f**) Oval. (**a**) t = 2.2 mm. (**b**) t = 2.2 mm. (**c**) t = 2.2 mm. (**d**) t = 0.8 mm. (**e**) t = 3.0 mm. (**f**) t = 2.5 mm.

**Figure 10 materials-15-00805-f010:**
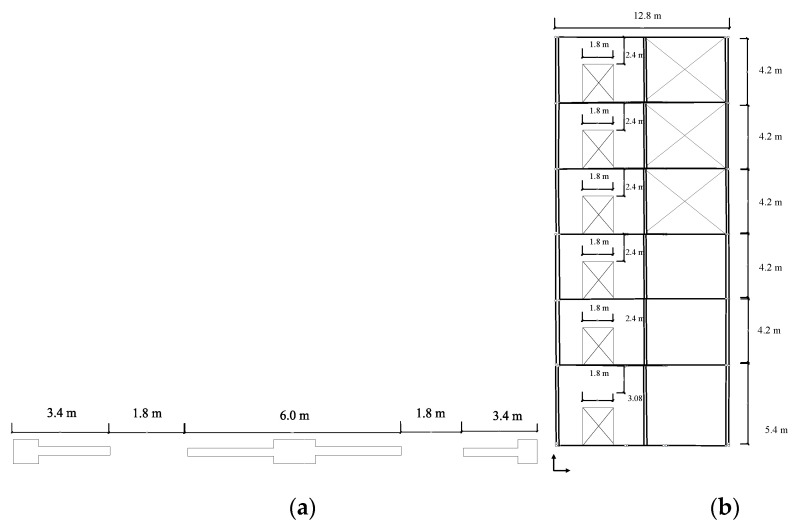
The coupled shear wall and implementation of MRLs (**a**) Top view (**b**) Plan view.

**Figure 11 materials-15-00805-f011:**
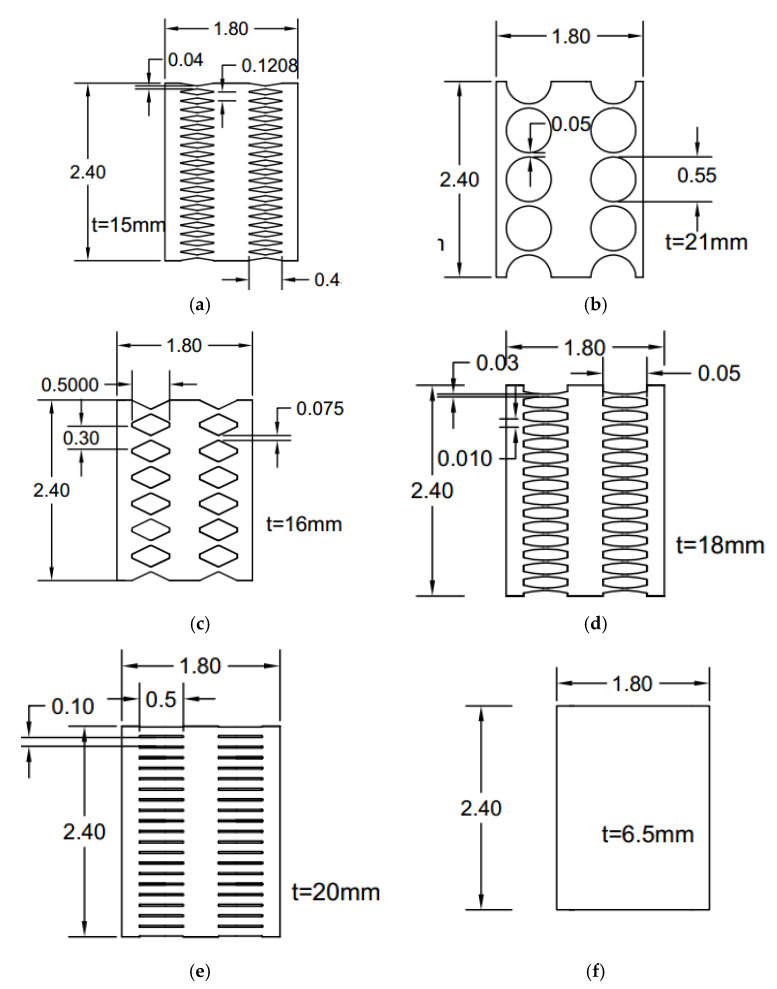
The multiple rows of links (**a**) FBF (**b**) Circle (**c**) SBF (**d**) Oval (**e**) Solid (**f**) Straight.

**Figure 12 materials-15-00805-f012:**
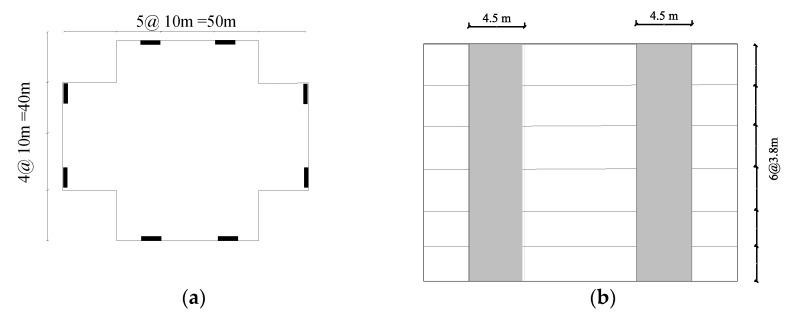
The steel shear wall application and location (2012) (**a**) The plan view (**b**) The elevation view.

**Figure 13 materials-15-00805-f013:**
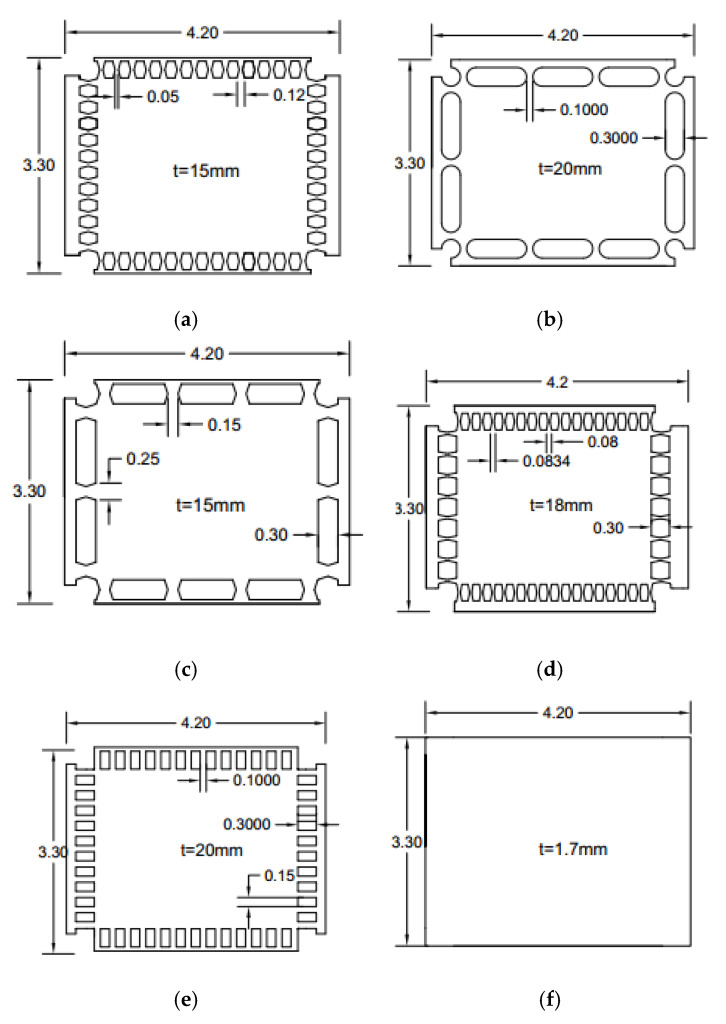
The conventional steel plate shear wall application (**a**) FBF (**b**) Circle (**c**) SBF (**d**) Oval (**e**) Straight (**f**) Solid.

**Figure 14 materials-15-00805-f014:**
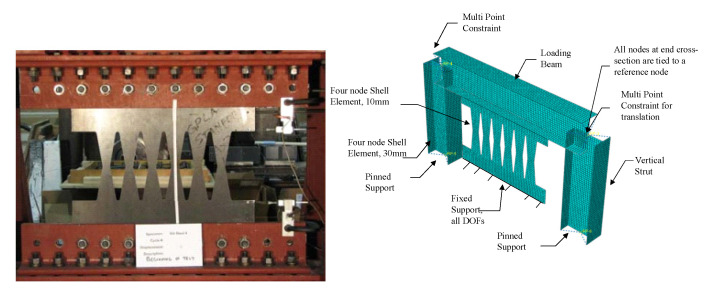
Specimen B10-36W done by Ma et al. [2] and the FE computational model description and details for FEM verification purposes.

**Figure 15 materials-15-00805-f015:**
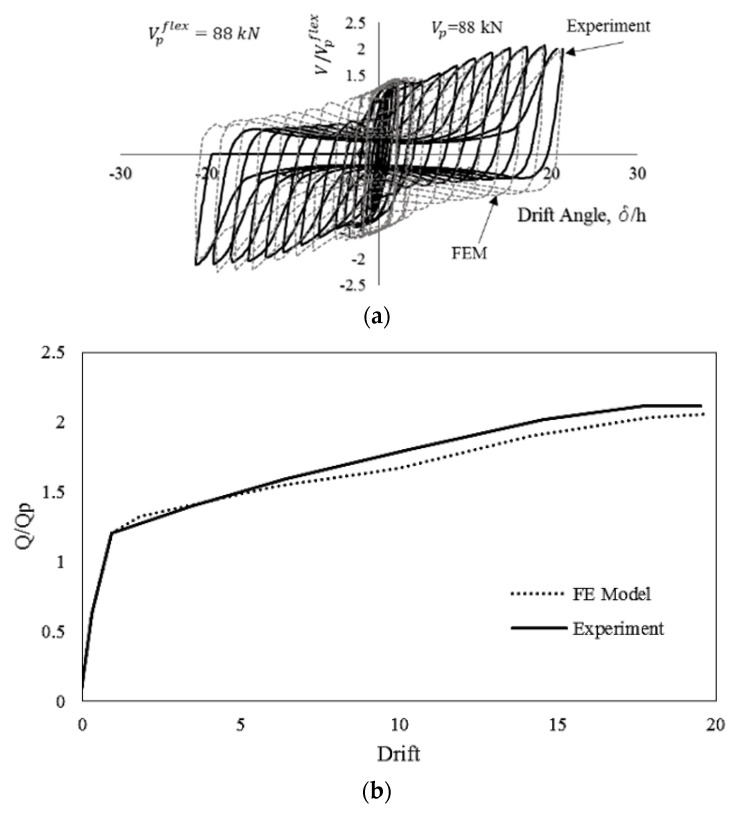
Verification of Finite Element Modeling with the aid of FE software under (**a**) cyclic behavior and (**b**) monotonic loading conditions.

**Figure 16 materials-15-00805-f016:**
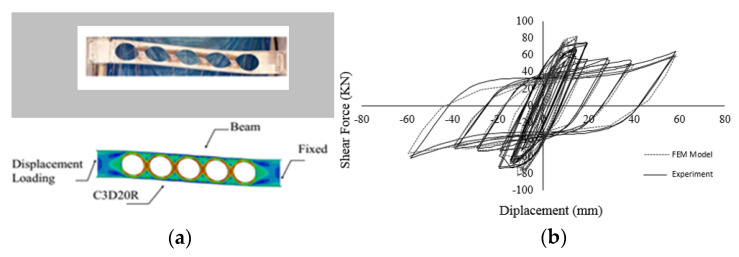
Verification of the second finite laboratory test (**a**) Test specimen and computational model (**b**) Load vs. deformation hysteretic response.

**Figure 17 materials-15-00805-f017:**
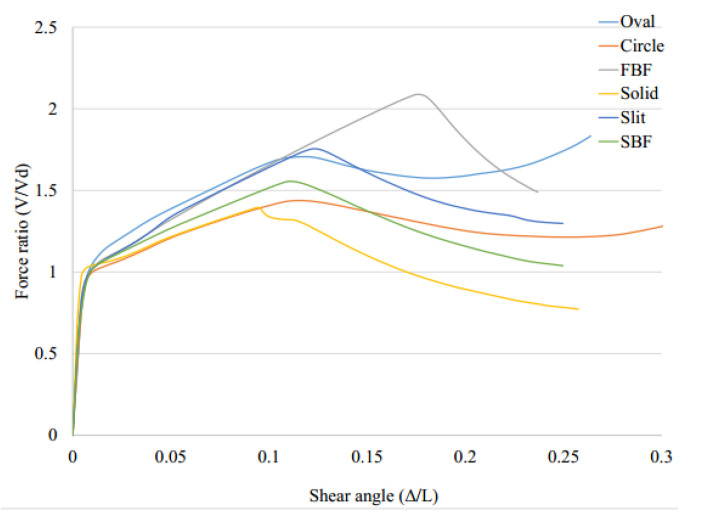
Pushover analyses curves for various innovative SRL fuse systems.

**Figure 18 materials-15-00805-f018:**
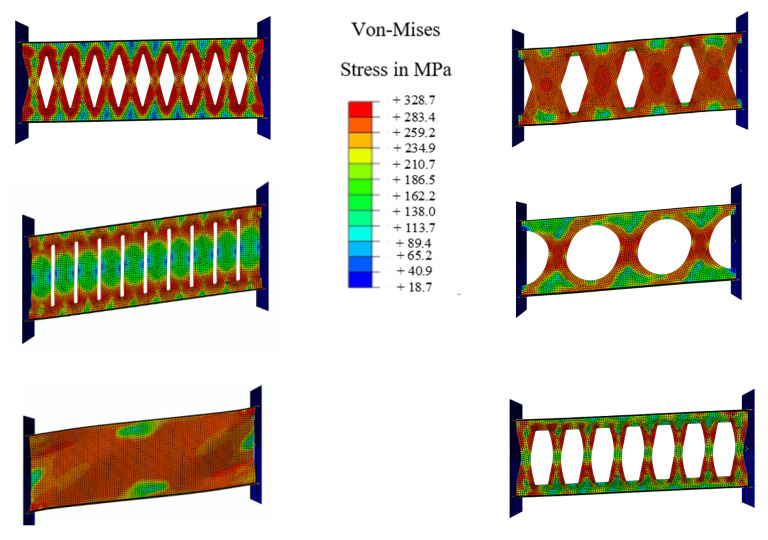
The Von-Mises stress, and stress concentration areas for SRLs at 0.08 drift ratio.

**Figure 19 materials-15-00805-f019:**
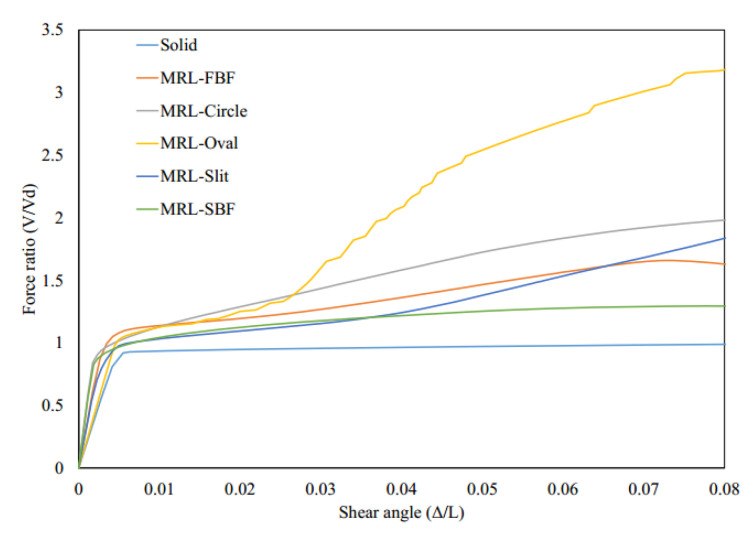
Pushover analyses curves for various innovative MRL fuse systems.

**Figure 20 materials-15-00805-f020:**
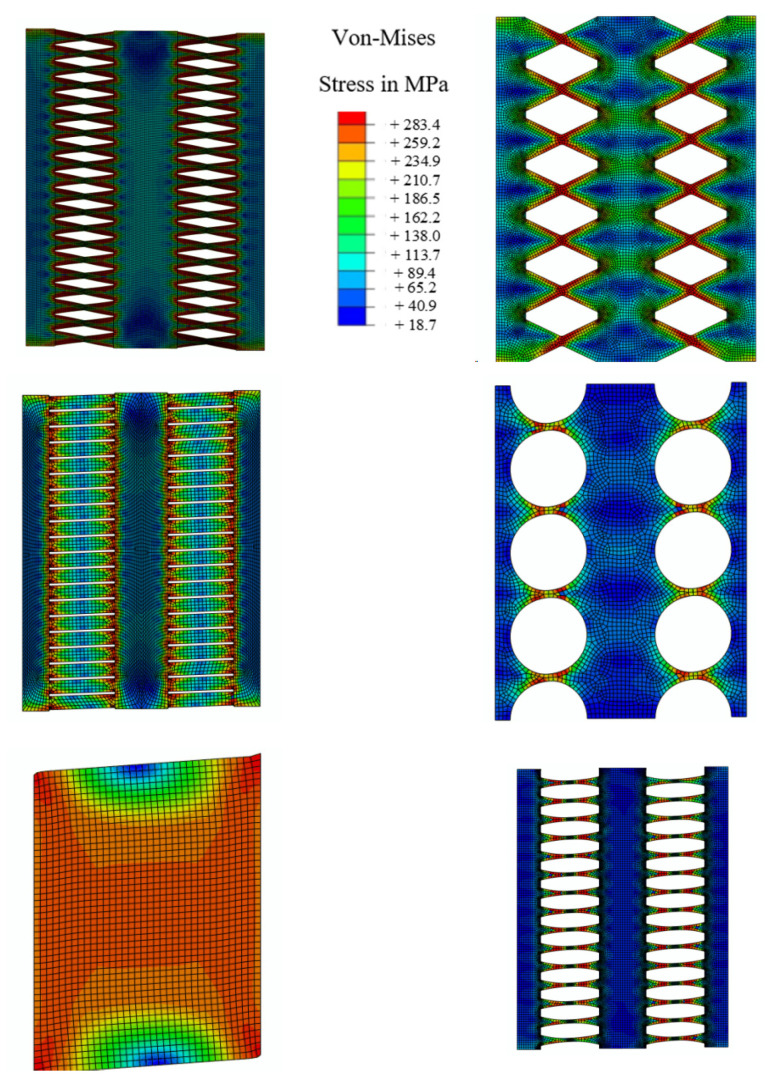
Von-Mises stress for MRLs.

**Figure 21 materials-15-00805-f021:**
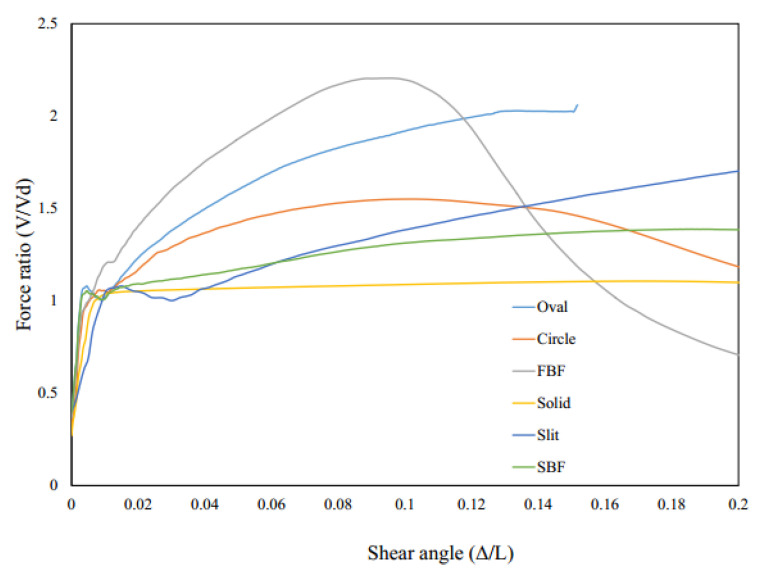
Pushover analyses curves for various innovative PRL fuse systems.

**Figure 22 materials-15-00805-f022:**
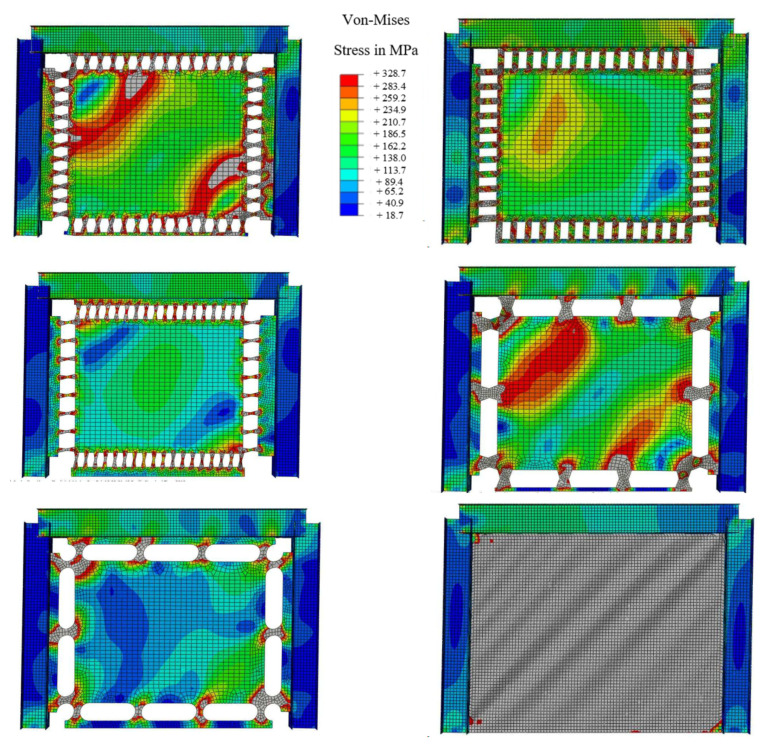
Von-Mises stress distribution for PRLs.

**Figure 23 materials-15-00805-f023:**
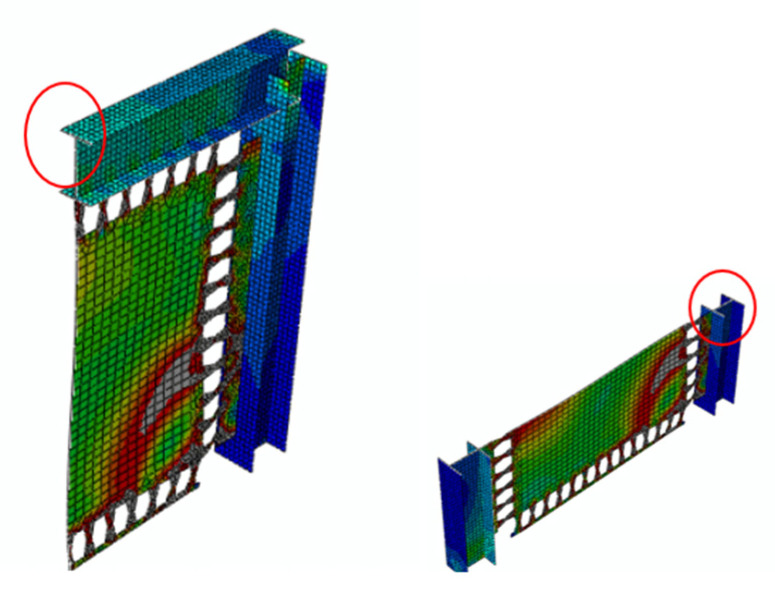
The indication of the mid-point area used for moment assessment.

**Table 1 materials-15-00805-t001:** The over strength factor established for butterfly-shaped dampers.

BF Links		b/L	Over Strength Ω
a/b	0.1	0.1	4.1
		0.2	3.3
		0.3	2.8
		0.4	1.8
	0.33	0.1	2.3
		0.2	1.65
		0.3	1.35
		0.4	1.3
	0.75	0.1	4.13
		0.2	3.18
		0.3	2.51
		0.4	1.95
	1	0.1	4.35
		0.2	3.35
		0.3	2.75
		0.4	2.45

**Table 2 materials-15-00805-t002:** The computational results for SRLs.

Output	Oval	Circle	FBF	Simple	Straight	SBF
PEEQ at 0.08 (−)	0.22	0.25	0.28	0.68	0.61	0.26
displacement, Dy (m)	0.012	0.008	0.0089	0.01	0.009	0.01
displacement, Dm (m)	0.13	0.115	0.21	0.115	0.14	0.135
Displacement ratio (−)	10.8	14.4	23.6	11.5	15.6	13.5
Ultimate Strength (kN)	908	862	1037	767	802	808
Yielding Strength (kN)	561	582	496	577	450	520
Over strength (−)	1.62	1.48	2.09	1.33	1.78	1.55
Stiffness (kN/m)	46,750	72,750	55,730	57,700	50,000	52,000

**Table 3 materials-15-00805-t003:** The mode of behavior for SRLs up to at 0.08 drift ratio.

Performance investigation	Oval	End of dampers elements excessive plasticity are obtained with minor buckling.
	Circle	Middle section of the dampers yields, and excessive rotation occured are the end section
	FBF	End elements start to yield by 0.08 drift ratio with uniform stress distribution.
	Simple	The buckling occurred, and the tension Field Action (TFA) is observed clearly.
	Straight	End of dampers are subject to excessive rotation The plasticity and fracture potentials have been indicated.
	SBF	The dampers are yielded in shear at the middle, with minor buckling.

**Table 4 materials-15-00805-t004:** The computational results for MRLs.

	Oval	Circle	FBF	Simple	Straight	SBF
PEEQ at 0.03 (−)	0.055	0.084	0.044	0.14	0.09	0.09
PEEQ 0.08 (−)	0.065	0.137	0.12	0.47	0.51	0.11
Displacement, Dy (m)	0.011	0.0084	0.011	0.0078	0.008	0.0056
Displacement, Dm (m)	0.21	0.22	0.185	0.19	0.38	0.197
Displacement ratio (−)	19.1	26.2	16.8	24.4	47.5	35.2
Ultimate Strength (kN)	4744	3073	2927	2505	7627	3476
Yielding Strength (kN)	1474	1481	1954	2284	2173	2354
Over strength (−)	3.22	2.07	1.50	1.10	3.51	1.48
Stiffness (kN/m)	134,076	176,354	177,673	292,895	271,680	420,443

**Table 5 materials-15-00805-t005:** The mode of behavior for MRLs up to at 0.08 drift ratio.

Performance investigation	Oval	The uniform yielding occurs with the length of the damper. Buckling is prevent and stress has been uniformly distributed; the plastic strain are low; therefore, the fraction prevention is occurred.
	Circle	Majority of the yielding occurs at the middle section. The buckling and excessive rotation did not happen
	FBF	The flexural limit state is clear and the stresses are uniformly distributed and excessive rotation at the middle is observed.
	Simple	The tension field action has occurred and buckling was clear.
	Straight	The end elements are yielded and the plastic strain at the end of the damper are high. Therefore, fracture potential is high.
	SBF	At the middle yielding occurred and buckling as the subsequent limit state occurs.

**Table 6 materials-15-00805-t006:** The computational results for PRLs.

	Oval	Circle	FBF	Simple	Straight	SBF
PEEQ at 0.02 (−)	0.24	0.11	0.40	0.045	0.3	0.21
Displacement, Dy (m)	0.0174	0.027	0.0162	0.0348	0.0099	0.023
Displacement, Dm (m)	0.478	0.349	0.332	0.61	0.468	0.62
Displacement ratio (−)	27.47	12.93	20.49	17.53	47.27	26.96
Ultimate Strength (kN)	2916	2232	3158	1591	2915	1998
Yielding Strength (kN)	1526	1525	1435	1497	1503	1476
Over strength (−)	1.91	1.46	2.20	1.06	1.94	1.35
Stiffness (kN/m)	87,720	56,483	88,565	43,009	151,818	64,156

**Table 7 materials-15-00805-t007:** The post-processing results for PRLs.

Performance investigation	Oval	The corner dampers undergo high buckling without the frame damaging from the plastic concentration.
	Circle	The top corner damper at left experiences high inelasticity stress concentration.
	FBF	The bottom left-hand side and right-hand side dampers undergo elongating and shortening.
	Simple	Tension field action is occured and high demands on the boudary elements are determiend.
	Straight	The whole panel initiates to buckle. The hinges are concentrated at the ends of each dampers.
	SBF	The middle part is subjected to rotational elements without occurrence of early buckling.

**Table 8 materials-15-00805-t008:** The demand moment forces captured at the middle point of beam and column.

Type	M (kN·m) at 0.02 Drift	
	Beam	Column
Straight	447	125
Simple	373	273
Oval	378	50
Circle	307	85
SBF	373	72
FBF	313	76
Mp (kN·m)	1288	2566.3

## Data Availability

The data presented in this study are available on request from the corresponding author.

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
