# Peer review of "Innovative Structural Fuse Systems for Various Prototype Applications"

_materials, 2022, doi:10.3390/ma15030805_

Round 1

Reviewer 1 Report

The article "Innovative Structural Fuse Systems for Various Prototype Ap-2 plications" is interesting. However, there is need to revised carefully before publication.

  1. There is need to quantitatively input major results in abstract.
  2. The introduction is not well written. The need and importance of this study is not clear.

There is need to summarize existing finite element analysis studies. Please consider few articles for references.

Gan, Y. (2000). Bond stress and slip modeling in nonlinear finite element analysis of reinforced concrete structures. University of Toronto.

Raza, A., Shah, S. A. R., Alhazmi, H., Abrar, M., & Razzaq, S. (2021). Strength Profile Pattern of FRP-Reinforced Concrete Structures: A Performance Analysis through Finite Element Analysis and Empirical Modeling Technique. Polymers13(8), 1265.

Chaimahawan, P., Suparp, S., Joyklad, P., & Hussain, Q. (2021). Finite Element Analysis of Reinforced Concrete Pile Cap using ATENA. Latin American Journal of Solids and Structures18.

3. Research methodology is not clear as well.

4. Page 5. It is concluded that, the uniform yielding along the damper length method indicates 150 that the width of the hourglass-shaped damper, w(z), should be corresponding to the square root of z, while the “same curvature” method indicates that b  is recommended to align with the cube root of z. The sentence is too long and meaning is not clear at all. Another thing, who concluded this and based on what?

5. Why authors do mention many itmes "It is noted"......

6. Figure 9, the dimension style is extremly large as compared to the figure.

7. There is need to explain material models adopted in the finite element analysis.

8. Results are not explained well and also there is no comparison with the previous studies.

9. Error! Reference source not found.......what is this? The authors are suggested to carefully read the paper before submission.

10. Conclusions should be to the point.

11. There are some unneceessary figures. 

Author Response

Innovative Structural Fuse Systems for Various Prototype Applications by Alireza Farzampour

Materials

Materials-1505091

Response to Reviewer Comments

Please find attached a second revised version of our manuscript “Innovative Structural Fuse Systems for Various Prototype Applications”, which we would like to revise-resubmit after major revision for publication as a Regular paper in buildings.

Your comments and those of the reviewers were highly insightful and enabled us to greatly improve the quality of our manuscript. In the following pages are our detailed and point-by-point responses to each of the comments of the reviewers as well as your own comments.

Revisions in the text are shown using yellow highlight for additions. In accordance with reviewers’ suggestions, we have revised following items. Please note since several comments form different reviewers are related to each other all comments are attached within single document.

We hope that the revisions in the manuscript and our accompanying responses will be sufficient to make our manuscript suitable for publication in buildings. We shall look forward to hearing from you at your earliest convenience.

Yours sincerely,

Alireza Farzampour, PhD, EIT

Government, USA

MDPI Materials Special Issue Editor

Reviewer 2 Report

The paper illustrates a comparison between different methodologies 
to desing shear dampers for structural applications. The idea is 
good but the paper is written really bad. It was very difficult, for 
the reviewer, to follow what the work aims to do. In this actual form,
the manuscript cannot be accepted for a possible pubblication. Notwithstanding, some maior and minor issues have been raised to indicate a reasonable path to improve the manuscript.

1) Page 1, line 41: "Figure" is repeated twice.

2) Figure 1: it could better be organized, for example could be highlighted 
the types where the mechanism is more influence than the shape and viceversa.

3) The introduction could be strengthened introducing some pioneeristic paper related to the dissipation. The english writing should be revised (some typing errors present).

4) Figure 2 and 3 are very poor. They should be strongly improved. Bending axis are without indication (x or y). Looking the Figure 2 is not even intuitively possible to understand what is "the concept of uniform yielding"....please try to improve as much as possible.

5) Sometimes, along the manuscript is found: "Error! Reference source not found ".

6) The variable w(z) has not been indicated in any figures.

7) In the beginning of paragraph 3, SEAOC and EBRF are acronyms never introduced before, please verify.

8) Figure 9 is organized in a very bad way. Please improve: sections and quotated measures (size, characters, overlapping).

9) Also the figure 10 is incomprehensible: improve indication of the axes and coupled sear wall. What does it mean "MRLs"?

10) For figure 11 are valid the same observations of figure 9.

11) Improve the definition of figure 13 and try to equal all choices 
(character and its dimension). Moreover, what does it mean in the figure 13f 
"straight"?

12) Improve the definition of the figures 14 and 15, moreover they have 
the same caption. Improve and correct.

13) The three legends illustrated in the Figure 18 are equal. Maybe, only one 
legend could be used trying to better organize the figure. Same observation 
for the Figures 21 and 23.

14) Try to make more robust and convincing the conclusion.

15) Since the auhtor is unique, the description of the "Author Controbutions"
is useless. 

Author Response

(The authors gave the same response as above.)

Reviewer 3 Report

The article is well set up and clear in its objectives. However, some of the figures are unclear and in others the captions could be improved.

Many of the references to figures and equations in the article are missing and the message «Error! Reference source not found.» appears in the text. This makes it arduous to comprehend some parts of the article, since it is first necessary to identify which figure, table or equation is being referred to.

More specific notes are reported in the attached file.

Author Response

(The authors gave the same response as above.)

Round 2

Reviewer 1 Report

Accept in present form.

Author Response

Thank you.  The comments are attached. 

Reviewer 2 Report

The actual version of the paper has been sufficiently improved following the observations raised by the reviewer. The only suggestion is a re-reading of the manuscript to correct the style of the english writing.

Author Response

Thank you. 
